# Selective Isolation of Liver-Derived Extracellular Vesicles Redefines Performance of miRNA Biomarkers for Non-Alcoholic Fatty Liver Disease

**DOI:** 10.3390/biomedicines10010195

**Published:** 2022-01-17

**Authors:** Lauren A. Newman, Zivile Useckaite, Jillian Johnson, Michael J. Sorich, Ashley M. Hopkins, Andrew Rowland

**Affiliations:** 1College of Medicine and Public Health, Flinders University, Adelaide, SA 5042, Australia; lauren.newman@flinders.edu.au (L.A.N.); zivile.useckaite@flinders.edu.au (Z.U.); michael.sorich@flinders.edu.au (M.J.S.); ashley.hopkins@flinders.edu.au (A.M.H.); 2Early Clinical Development, Pfizer Global Research and Development, Groton, CT 06340, USA; jillian.johnson@pfizer.com

**Keywords:** microRNA biomarkers, extracellular vesicles, liver-specific isolation, non-alcoholic fatty liver disease, non-alcoholic steatohepatitis

## Abstract

Non-alcoholic fatty liver disease (NAFLD) is the most common chronic liver disease. Definitive diagnosis of the progressive form, non-alcoholic steatohepatitis (NASH), requires liver biopsy, which is highly invasive and unsuited to early disease or tracking changes. Inadequate performance of current minimally invasive tools is a critical barrier to managing NAFLD burden. Altered circulating miRNA profiles show potential for minimally invasive tracking of NAFLD. The selective isolation of the circulating extracellular vesicle subset that originates from hepatocytes presents an important opportunity for improving the performance of miRNA biomarkers of liver disease. The expressions of miR-122, -192, and -128-3p were quantified in total cell-free RNA, global EVs, and liver-specific EVs from control, NAFL, and NASH subjects. In ASGR1+ EVs, each miR biomarker trended positively with disease severity and expression was significantly higher in NASH subjects compared with controls. The c-statistic defining the performance of ASGR1+ EV derived miRNAs was invariably >0.78. This trend was not observed in the alternative sources. This study demonstrates the capacity for liver-specific isolation to transform the performance of EV-derived miRNA biomarkers for NAFLD, robustly distinguishing patients with NAFL and NASH.

## 1. Introduction

Non-alcoholic fatty liver disease (NAFLD) is the most common chronic liver disease, affecting up to a third of the global population [1]. The disease manifests on a spectrum of severity, with most individuals presenting with benign hepatic fat accumulation (non-alcoholic fatty liver; NAFL) and approximately 30% exhibiting a more severe form known as non-alcoholic steatohepatitis (NASH) [2]. While insulin resistance, obesity, and other features of the metabolic syndrome are commonly associated with NAFLD, the aetiology of the disease remains largely unknown, particularly with respect to its progression to NASH [3,4]. Treatment guidelines consistently identify early detection and intervention as key to improving the clinical outcomes and to reducing the burden of NAFLD [5,6]. NAFLD is an independent mortality risk factor, with all-cause mortality on average, 11.7% higher in individuals with NAFLD compared with those without. The impact on mortality among individuals with NAFLD is proportional to disease severity and ranges from 8.3% for NAFL up to 18.4% for NASH and fibrosis [7].

NAFLD is diagnosed in individuals that exhibit fatty changes in more than 5% of hepatocytes where other causes of steatosis have been excluded [8,9]. Liver biopsy is the gold standard for NAFLD staging and the only approach to reliably defining fibrosis [10]. The key limitation is that liver biopsy is an invasive procedure that carries a substantial risk of complication, including bleeding and infection. This precludes biopsy in low-risk patients and limits utility in high-risk patients, as the procedure can only be performed once every two years. Additionally, given the sporadic infiltration of NAFLD and the limited tissue sample achieved with a needle biopsy, there is a substantial risk of inaccurate diagnosis that underestimates disease stage [9]. Given these limitations, a range of minimally invasive approaches to diagnose and stage NAFLD have been proposed. These approaches are typically based on factors such as serum biomarkers, body composition, comorbid diseases, and abdominal imaging [8,9,10]. Different combinations of these factors produce the fatty liver index, hepatic steatosis index, and liver fat scores, which are used as screening tools for NAFLD. While strong predictive performance has been reported for some indices, there is insufficient robustness to facilitate translation to routine clinical care, particularly for mild and early disease [11]. Indeed, in 2019, the American Association for the Study of Liver Diseases identified the inadequate performance of these tools as a critical barrier to the effective treatment of patients with NAFLD [6].

MicroRNAs (miRNA) are small non-coding RNA that have been shown to reflect disease-associated changes across numerous pathological conditions [12]. Altered expression of miRNAs that mediate pathways involved in lipid metabolism, inflammatory activation, and the development of fibrosis have been observed in tissue from individuals with NAFLD. The abundance of these miRNAs in blood has been postulated as potential biomarkers used to diagnosis and track NAFLD [13,14]. This approach potentially confers several advantages over liver biopsy. In addition to mitigating the risks of tissue biopsy, circulating miRNA analysis facilitates longitudinal evaluation of disease and potentially a more robust overview of disease stage. The stability of cell-free (cf) miRNA in blood is attributable to protection by RNA binding proteins, such as argonaute 2 (Ago2), high-density lipoproteins (HDL), or encapsulation within extracellular vesicles (EVs) [15].

EVs are small membrane-bound particles released by virtually all cell types into various biological fluids including blood, urine, and cerebrospinal fluid [16,17]. EVs carry an array of nucleic acid (including miRNA), protein, and lipid cargo derived from the parent cell [17,18]. It has been proposed that isolating EVs from biological fluids may improve the fidelity of miRNA biomarker analyses as the EV fraction provides a source of miRNA that is selectively packaged in a more disease-specific manner [19]. Additionally, circulating EVs express cell surface markers from the originating tissue. Thus, in contrast with circulating miRNA bound to Ago2 and HDL, it is theoretically possible to selectively isolate EV miRNA that originates from a specific tissue. It has been proposed [20] that analysis of EVs selectively isolated from an afflicted organ provide the most informative description of biomarker expression, as this fraction is less affected by ‘noise’ associated with non-specific fluctuations in global EV and, in the case of miRNA, total circulating expression.

To date, studies of cell-free miRNA in NAFLD have focussed broadly on expression in whole plasma and serum. When simplified to a dichotomous analysis of healthy versus NAFLD, moderate diagnostic performance in terms of discriminating these two groups has been reported for a number of miRNAs both as individual markers and panels [13,21]. Despite intriguing preliminary results, the consistency of results from miRNA profiling studies remains insufficient to be applied in practice [15]. Furthermore, this dichotomous grouping of healthy versus disease limits meaningful interpretation of an individual’s NAFLD risk, which differs substantially with disease severity. For the current study, three representative miRNAs were selected based on their reported liver specificity (miR-122), associations with steatosis and fibrosis (miR-122, -192, and -128-3p) and association with inflammation (miR-192). This study sought to identify trends in the expressions of these three miRNAs; to determine whether the sequential refinement of the source from which biomarkers are quantified; and to improve their predictive power with respect to differentiating NAFL, NASH, and control subjects.

## 2. Materials and Methods

### 2.1. Study Population and Blood Samples

Clinically annotated K_2_EDTA plasma from NAFLD patients (NAFL *n* = 8; biopsy-proven NASH *n* = 6) and healthy donors (*n* = 14) matched for age and sex were purchased from Discovery Life Sciences (Hunstville, AL, USA). The inclusion criteria included a clinical diagnosis of NAFL or NASH by a physician, and the exclusion criterion was the presence of a viral disease. The samples were aliquoted for EV isolation and total RNA analysis according to the study workflow depicted in Figure 1 and stored at −80 °C. All analyses were performed on all the patients, unless otherwise indicated. The demographic data describing each of the three study populations are presented in Table 1. All relevant data have been submitted to the EV-TRACK knowledgebase (ID EV210168) [22].

### 2.2. Isolation of Extracellular Vesicles

#### 2.2.1. Size Exclusion Chromatography

Global EVs were isolated using qEV2 70 nm size exclusion chromatography (SEC) columns (iZon Science, Christchurch, New Zealand). Prior to performing the isolations, the columns were equilibrated to room temperature and washed with 10 mL of 0.2 µm filtered phosphate-buffered saline (PBS). Plasma (1700 µL) was diluted up to 2 mL with PBS, loaded into the sample reservoir, and allowed to completely pass into the column, before eluting with PBS. The first six fractions eluted from the column were discarded, and vesicles were collected as a pool of fractions 7 to 11 (10 mL). Pooled vesicle fractions were concentrated to 400 µL using Amicon Ultra-15 centrifuge 30 kDa filters (Millipore-Sigma, Bedford, MA, USA) pre-conditioned with PBS and stored at −80 °C until analysis.

#### 2.2.2. Liver-Specific EV Immunoprecipitation

EVs specifically derived from the liver were separated from global EV isolates following a previously published protocol [23]. Briefly, 1.5 mg of Dynabeads M280 streptavidin magnetic beads (Cat#11205D, Thermo Fisher Scientific, Waltham, MA, USA) were pre-washed with PBS and incubated with 15 µg of biotinylated anti-asialoglycoprotein receptor 1 (ASGR1) polyclonal antibody (Cat#LS-C685544, 0.5 mg/mL, Sapphire Bioscience, Redfern, NSW, Australia) for 30 min at RT with gentle agitation. Antibody-conjugated beads were separated using a DynaMag-2 magnet (Thermo Fisher Scientific, Waltham, MA, USA), washed with 0.1% bovine serum albumin in PBS, and resuspended in PBS. Concentrated qEV70 vesicles (150 µL) were added to antibody-coated beads and incubated for 24 h at 4 °C on a rotating mixer. ASGR1+ EVs bound to the antibody-bead complexes were separated on the magnet, washed, and resuspended in PBS.

### 2.3. Transmission Electron Microscopy

Samples were prepared based on a previously published protocol [17]. Briefly, 5 μL of the EV sample in filtered PBS was placed on carbon-coated grids for 4 min (Ted-Pella B 300M, Mason Technology, Dublin, Ireland). Grids were washed for 15 s with 0.2 μm filtered PBS at room temperature and were contrasted with 2% uranyl acetate (3 min at room temperature), washed once, and examined by FEI TECNAI Spirit G2 TEM (Thermo Fisher Scientific, Waltham, MA, USA).

### 2.4. Nanoparticle Tracking Analysis

Nanoparticle tracking analysis (NTA) was performed to quantify particle concentration and size distribution in EV samples using a NanoSight NS300 (Malvern Analytical, Malvern, UK). Samples were diluted between 1:500 and 1:2000 in PBS. Ten 60-s videos were captured at camera level 14 with continuous sample flow (flow rate 100), and the videos were analysed at a detection threshold of 5 using NTA 3.4 software.

### 2.5. Total Protein Concentration

EVs were lysed by the addition of a RIPA buffer at a ratio of 1:1, incubated on ice for 25 min and centrifuged at 10,000× *g* for 10 min at 4 °C. Total protein concentration was determined using Pierce MicroBCA Protein Assay following manufacturer’s instructions (Thermo Fisher Scientific, Waltham, MA, USA).

### 2.6. Peptide Digestion

EVs (50 µL) were diluted up to 100 µL in PBS, vortexed for 10 min using a MixMate sample mixer (Eppendorf South Pacific, North Ryde, NSW, Australia), and then lysed by freezing and thawing for three times. Lysed EVs were combined with 50 µL of ammonium bicarbonate (pH 7.8) and incubated with dithiothreitol (DTT; 12.5 mM) for 90 min at 60 °C. Samples were cooled to room temperature prior to the addition of iodoacetamide (IAA; 23.5 mM) and incubation for 60 min at 37 °C. Trypsin Gold was then added to the EV protein samples in a ratio of 1:40 and incubated for 18 h at 37 °C. Digests were terminated by the addition of 20 µL of formic acid (10% *v*/*v*), then centrifuged at 16,000× *g* for 10 min at 4 °C. A 100 µL aliquot of the resulting supernatant was combined with SIL peptide standards (25–2500 pM cocktail; Vivitide, Gardner, MA, USA) and a 5 µL aliquot was injected for analysis by LC–MS/MS.

### 2.7. LC–MS Peptide Analysis

Chromatographic separation of peptide analytes was performed on an Agilent Advance Bio Peptide Map column (100 × 2.1 mm, 2.7 µm) using an Agilent 1290 Infinity II liquid chromatography system. The temperatures of the column and sample compartment were maintained at 30 and 4 °C, respectively. Separation was achieved by gradient elution with a flow rate of 0.2 mL/min. The mobile phase consisted of 0.1% formic acid in water (mobile phase A) and 0.1% formic acid in acetonitrile (mobile phase B). The proportion of mobile phase B was held at 10% for 2 min and then increased to 60% over 13 min, before returning to 10% for 1 min. The column was re-equilibrated for 30 s, and the total run time was 16.5 min. Column eluant was monitored by mass spectrometry using an Agilent 6495B Triple Quadrupole mass spectrometer operating in positive electron spray (ESI +) mode. Multiple reaction monitoring (MRM) was performed with one quantifier and one qualifier ion transition for each peptide. The identities of the endogenous peptides were confirmed by comparing retention time and quantifier/qualifier transition ratios to respective SIL peptide standards and relative abundance determined by quantifier ion peak area.

### 2.8. RNA Isolation

Total RNA was isolated from plasma and EV samples using TRIzol LS^TM^ Reagent (Thermo Fisher Scientific, Waltham, MA, USA) as per manufacturer’s instructions with some modifications. Briefly, 750 µL TRIzol LS was added to 200 µL of plasma and 100 µL of EVs (diluted up to 200 µL in RNase-free water). Samples were spiked with 2.5 femtomoles of cel-miR-54 mirVana mimic (MC10279, Thermo Fisher Scientific, Waltham, MA, USA) to normalise for technical variability in RNA extraction and RT-qPCR efficiency. This exogenous control was employed in the absence of established endogenous genes for normalisation of plasma or EV-derived miRNA RT-qPCR data [24,25]. Isopropyl RNA precipitation was facilitated by addition of 40 µg of RNase-free glycogen (Thermo Fisher Scientific, Waltham, MA, USA), and the RNA pellet was washed with ice cold 80% ethanol. RNA was resuspended in 30 µL of RNase-free water.

### 2.9. RT-qPCR

Equal volumes of RNA (5 µL) were reverse transcribed using the TaqMan microRNA Reverse Transcription Kit (Thermo Fisher Scientific, Waltham, MA, USA). Since the concentration of circulating cfRNA is usually below the limit of quantification for photometric or colorimetric techniques, equivalent mass could not be reliably determined. Thus, RNA input was based on a fixed volume rather than RNA mass (ng), as in previous studies [12,26,27]. Instead, alterations in miRNA levels in equivalent volumes of plasma or EV isolates were detected relative to the exogenous spike in. TaqMan Small RNA Assays (Thermo Fisher Scientific, Waltham, MA, USA) were used to carry out qPCR assays with primers specific to the miRNA of interest (hsa-miR-122 (002245), hsa-miR-192 (00491), miR-128a (002216), and cel-miR-54 (001361)) in a Rotor-Gene 3000 (Corbett Research, Sydney, NSW, Australia). Samples were assayed in duplicate and the same cycle threshold set across all runs.

### 2.10. Statistical Analysis

Cycle threshold (Ct) values derived from RT-PCR were used to calculate relative quantities (RQ) according to the following formula:$$\mathrm{RQ}=2^{-\left(\mathrm{mean}\;\mathrm{biomarker}\;\mathrm{Ct}\;-\;\mathrm{mean}\;\mathrm{spike}–\mathrm{in}\;\mathrm{Ct}\right)}$$

Statistical analysis was performed using GraphPad Prism software version 9.0 (San Diego, CA, USA). The data are presented as mean ± SD unless specified. Statistical comparisons between the NAFL, NASH, and control groups were performed using the Kruskal–Wallis test and Dunn’s test for multiple comparisons. Statistical significance was set at 0.05. A receiver operating characteristic (ROC) analysis was used to assess the diagnostic capacity of miRNA biomarkers between pairs of groups. Ordinal logistic regression was performed using R version 1.4 (Boston, MA, USA) to determine diagnostic value across the three groups.

## 3. Results

### 3.1. Isolation and Characterization of Circulating EVs from NAFLD and Control Subjects

The concentration and size of global EV particles isolated from NAFL, NASH, and control subjects was determined by NTA. An apparent higher mean EV concentration (±SD) of 4.17 × 10^11^ (±1.76 × 10^11^) particles/mL was observed in control subjects compared with NAFL (2.34 × 10^11^ (±9.03 × 10^10^) particles/mL) and NASH (2.73 × 10^11^ (±1.01 × 10^11^) particles/mL) subjects (Figure 2a, Table 2). However, given the marked within group heterogeneity, no statistically significant differences in EV concentration were detected between groups. Mean particle size, given in Table 2, also did not vary between control, NAFL, and NASH groups.

Global EVs were also analysed by TEM to assess sample composition and morphology. TEM images of EVs from NASH and control subjects isolated by qEV revealed characteristic EV morphology, size, and removal of non-vesicular contaminants (Figure 2b and Appendix A). While the EVs were predominantly of similar size between NASH and control subjects, the presence of a few larger EVs was noted in the control sample.

Established positive and negative EV protein markers were probed by targeted LC-MS peptide analysis [17]. Positive markers, as described by the Minimal Information for Studies of Extracellular Vesicles (MISEV) [28], include tetraspanins, CD81 (Class 1a), CD9 (Class 1b), and tumour susceptibility gene 101 (TSG101; Class 2a) and were detected in EV isolates from control, NAFL, and NASH subjects (Figure 2c). Only CD81 showed a significantly lower abundance in NAFL compared with control samples; however, total protein concentration did not differ between groups (Appendix A). Samples were also analysed for negative markers of matrix contamination and cellular debris; albumin (Class 3b) and Calnexin V (Class 4c). Minimal expression was observed for both negative markers; the mean abundance of albumin and calnexin in EV preparations was 0.21% and 0.94% of abundance in respective positive controls (plasma for albumin and human liver microsomes for calnexin).

Taken together, these data indicate that qEV size exclusion chromatography permits the isolation of pure EVs with characteristic molecular and physical properties from the circulation of patients with NAFLD and healthy controls. Characterisation of liver-specific EV isolates was performed with respect to particle concentration and ASGR1 abundance and is reported as Supplementary Data. Data are reported in accordance with MISEV guidance. EV-TRACK study identifier is EV210168.

### 3.2. Expression of Total Cell-Free, Global EV, and Liver-Specific EV miRNA Biomarkers

Expression of miRNA was quantified in total cfRNA, global EVs, and liver-specific EVs isolated by anti-asialoglycoprotein receptor 1 (ASGR1) immunoprecipitation. Relative quantities (RQ) for miR-122, -192, and -128-3p were calculated after normalisation to an exogenous spike-in (cel-miR-54) and compared between control, NAFL, and NASH subjects (Figure 3). The expressions of all three miRNA biomarkers were significantly greater in ASGR1+ EVs from NASH subjects compared with controls (*p* = 0.012 (miR-122), *p* = 0.013 (miR-192), and *p* = 0.032 (miR-128-3p)). Interestingly, this trend was not observed when miRNA was analysed from total cell-free or global EV sources. In global EVs, only miR-128-3p exhibited altered expression with significantly lower RQ in NAFL (*p* = 0.009) and NASH (*p* = 0.019) subjects compared with controls. The apparent alterations in expression were consistent irrespective of using total cell-free or global EV as the source of RNA (Figure 3c), indicating that the isolation of global EVs did not confer an appreciable benefit towards the capacity for miR-128-3p expression to distinguish the groups. Conversely, RQ for miR-122 and -192 did not vary between disease or control subjects in total cfRNA or global EV analysis. Overall, these data suggest that the selective analysis of miRNA biomarkers from liver-specific EVs has the potential to elucidate a useful trend in expression corresponding with disease stage, particularly for biomarkers with high tissue specificity such as the liver-specific miR-122 and, to a lesser extent, miR-192 and -128-3p.

### 3.3. The Proportion of Circulating miRNA Contained in EVs Changes with NAFLD

To further explore the distribution of miRNA biomarkers amongst circulating fractions, RQs were used to calculate the expression of miRNA biomarkers in global EVs as a percentage of total cfRNA and in ASGR1+ EVs as a percentage of global EVs (Table 3). It has been observed that RNA isolation from unfractionated samples results in substantially higher yield compared with EVs due to the contribution of other circulating miRNA complexes. To ensure sufficient sensitivity, the equivalent starting volume for EV RNA isolation was 2.5× that for total cfRNA and reported proportions were normalised accordingly. The results demonstrated that, in control samples, mean (±SD) percentage expression in global EVs/total RNA was no greater than 5.6% (±10.0%), suggesting that only a minor proportion of total cell-free RNA is found in EVs. Likewise, in control subjects, the mean percentage expression in ASGR1+/global EVs was similarly low. Interestingly however, there was a positive trend in proportional expression of all miRNA biomarkers with NAFLD. In NASH, 27.1% of vesicular miR-122 signal came from ASGR1+ EVs compared to just 2.4% in control subjects (*p* = 0.035) (Table 3, Figure 4a). Significantly greater ASGR1+/global EV expression was also observed for miR-128-3p between NASH and control subjects (*p* = 0.022) (Figure 4c), while a similar difference in miR-192 fell short of statistical significance (*p* = 0.067) (Figure 4b). While only a small fraction of each miRNA is released in EVs expressing ASGR1—so a larger fraction may be derived from cells other than hepatocytes—these data suggest that, in NAFLD, the contribution of the liver to circulating EV-derived miRNA increases.

### 3.4. Association between miRNA Expression and Disease Severity

Ordinal logistic regression was performed to evaluate associations between miRNA abundance and the probability of a subject being healthy (control), NAFL, or NASH. Significant associations between miRNA expression and group status were observed for all three miRNAs in liver-specific EVs (Figure 5). The C-statistic for miR-122 was 0.80 with an odds ratio (OR) of 0.60 (*p* = 0.004), indicating that, for every 1 unit decrease in ∆Ct, the subject was 40% more likely to be NASH than NAFL or NAFL than control. Similarly, the C-statistics for miR-192 and miR-128-3p were 0.84 and 0.78, with ORs of 0.51 (*p* = 0.005) and 0.65 (*p* = 0.016), respectively. The expressions of miR-122 and miR-192 in total cfRNA and global EVs demonstrated no significant association with group status. Robust predictive performance was observed for total cfRNA and global EV derived miR-128-3p, with C-statistics of 0.78 (*p* = 0.007) and 0.83 (*p* = 0.009), respectively. Of the miRNA sources, only liver-specific EVs demonstrated a consistent trend amongst all miRNAs across increasing disease state.

### 3.5. Capacity to Distinguish Subjects with Disease from Control

A receiver operator characteristic (ROC) analysis was undertaken to establish the performance of each miRNA marker in the current analysis in a manner consistent with prior studies, as summarised in our recent review [13] (Table 4). The abundance of total plasma RNA derived miR-192 and miR-128-3p, but not miR-122, robustly distinguished individuals with NAFLD from controls (AUC ≥ 0.714). Comparatively poor performance was observed for global EV-derived miRNAs, with only miR-128-3p (AUC = 0.888) distinguishing these groups. As per the ordinal logistic regression analysis, liver-specific EV-derived miRNAs demonstrated the strongest performance with respect to distinguishing individuals with NAFLD from controls. AUC values for liver-specific EV derived miRNAs were invariably >0.8. An extended ROC analysis defining the performance of miRNA marker with respect to distinguishing individual paired groups (i.e., control vs. NAFL, control vs. NASH, and NAFL vs. NASH) is reported in Appendix A.

## 4. Discussion

This study provides the first direct evidence supporting transformative improvement in the predictive performance of existing NAFLD miRNA biomarkers achieved by isolating liver-specific EVs. Specifically, it is demonstrated that following anti-ASGR1 immunoprecipitation, the expression of liver-specific EV-derived miR-122, -192, and -128-3p is significantly associated with disease severity. In all cases, there was a significant trend of greater miRNA expression in subjects with NASH compared with NAFL, and NAFL compared with control subjects (Figure 3). Notably, this trend was not observed with either total cell-free or global EV-derived RNA. Rather, an analysis from these two sources of RNA revealed a significant decrease in miR-128-3p expression with disease, while miR-122 and -192 were not altered in comparison with the controls. In all cases, liver-specific (ASGR1+) EV-derived miRNA biomarkers demonstrated strong capacity to predict subject status as the control, NAFL, or NASH (Figure 5) with c-statistics > 0.78. This performance was not observed with either total cfRNA or global EV-derived miRNAs. While the improved predictive performance associated with liver-specific isolation provides additional support for these markers reflecting a hepatic manifestation, the possibility that these markers may also be associated with more systemic comorbidities cannot currently be excluded.

Indeed, each of the miRNAs investigated here is abundant in liver tissue, especially miR-122, which accounts for about 70% of all miRNA expression in the liver [29]. However, miR-122 is also found in cardiac and skeletal muscles; miR-192 is abundant in the kidneys, intestine, and adipose tissue [30]; and miR-128-3p is expressed in numerous tissues types, including in the CNS [31] and adipose tissue [32]. miR-128-3p also acts as a tumour suppressor miR and is reported to be dysregulated in several cancers, including lung [33], colorectal [34], and hepatocellular [35,36] cancer. Fundamentally, circulating miRNAs lack tissue specificity; their presence and stability in biological fluids results from release by multiple cell types and localisation within EV, lipoprotein, and Ago protein complexes [37]. The mechanisms of miRNA export from cells are known to be differentially affected by disease states; accordingly, perturbation in the relative expression of circulating miRNAs in vesicular or non-vesicular compartments are anticipated [12,15]. Consistent with this phenomenon, it was observed that the fraction of total circulating miR-122 (which accounts for 70% of hepatic miRNA) contained within liver-specific EVs increased with disease stage. This observation further supports the hypothesis proposed by several recent studies that optimising the circulating miRNA source is a key step in translating miRNA biomarker analysis [12,15,19,38].

Jiang et al. [39] recently reported that, compared with total serum-derived miRNA, differences in EV-derived miRNAs, including miR-122 (ROC = 0.79), more effectively discriminated subjects with NAFLD from healthy controls. In contrast with Jiang, et al. [39], in the current study isolation of global EVs did not improve the predictive performance of miRNA biomarkers with respect to identifying individuals with NAFLD. It is plausible that differences in performance gains achieved by global EV isolation relate to differences in the composition of the NAFLD cohort (i.e., the proportion of individuals with NAFL versus NASH or the proportion of individuals with non-hepatic co-morbidities). While this study did not reproduce the improvement in predictive performance achieved by isolating global EVs, we did demonstrate that further refinement of the miRNA source by tissue-specific EV isolation markedly improved performance. This outcome is consistent with analyses demonstrating improved signal to noise for minimally invasive protein biomarkers in plasma for neuronal pathology [40,41,42,43] and cerebrovascular disease [44] with tissue-specific isolation. As per the current study, selectively enriching the biomarker source was consistently found to increase sensitivity and specificity of analyses. One study further showed that the expression of miR-382 in intestine-specific EVs, but not global EVs, could predict the functional activity of human breast cancer resistance proteins [45]. Furthermore, we recently applied the liver-specific EV isolation method used here to describe the induction of hepatic drug metabolising enzymes and transporters resulting from metabolic drug interactions and pregnancy [23]. Collectively, these reports highlight the emerging potential to leverage tissue-specific EVs as minimally invasive liquid biopsy platform.

The use of anti-ASGR1 immunoprecipitation facilitated the analysis of EV encapsulated miRNA biomarkers released only by hepatocytes. Given the reported high abundance of miR-122 in hepatocytes, a high proportional expression of miR-122 in ASGR1+ EV relative to global EV-derived miRNA was anticipated. However, in EVs isolated from control subjects, the fraction of EV-derived miR-122 recovered from ASGR1+ EVs was strikingly low (2.4% of global EV derived miR-122 and 0.13% of total circulating miR-122). This low expression of miRNA in ASGR1+ EVs likely reflects the low fraction of circulating EVs that are derived from hepatocytes. The substantial expression of miR-122 from non-hepatic sources emphasises the potential for trivial changes in non-hepatocyte-derived miRNAs to markedly impact NAFLD biomarker analysis in the absence of liver-specific EV isolation. Interestingly, the fraction of global EV and cfRNA-derived miR-122, -192, and -128-3p recovered in ASGR1+ EVs was markedly higher in NAFL and NASH subjects compared with controls, despite the circulating EV count remaining unchanged. This marked increase in ASGR1+ EV-derived miRNA expression is consistent with the reported roles of these miRNAs in the inflammatory processes and remodelling associated with NAFLD and supports the biological rationale for their use as biomarkers for this disease. It is worth noting that, while the expression of miR-122 and -128-3p was significantly increased in NAFL and NASH compared with control subjects (Figure 4), the proportion of these miRNAs in ASGR1+ EV still accounted for <30% of global EV expression and <2.5% of total cfRNA expression for each miRNA.

It is conceivable that a population of miRNA-containing hepatic EVs does not express ASGR1 on their surface and hence are not captured by the immunoprecipitation method applied in the current analysis. Alternatively, various extra-hepatic tissues expressing these miRNAs may contribute more profoundly to their abundance in circulating global EVs. NAFLD is notably the hepatic manifestation of a broader multi-system disease associated with systemic inflammation and metabolic disturbances in several organs. It is likely that perturbations in other organs may contribute to the observed differences in global EV cargo derived from subjects with NAFL and NASH. This explanation further emphasises the potential benefits of selectively isolating liver-specific EVs for biomarker analyses. The data presented in a recent study by Povero et al. [46] indicated that EVs expressing ASGR1 comprise approximately 20% of the circulating global pool and that the number of hepatocyte-derived EVs increases with severity of NASH. However, in the limited sample size analysed here, no differences in global EV abundance were observed between groups. In this context, it is not clear if the increased proportional expression of miRNA with disease was due to greater number of ASGR1+ EVs or increased packaging and export of miRNA biomarkers into vesicles.

The observed increase in ASGR1+ EV miRNA expression is consistent with a large body of work describing the dysregulated miRNA profile in NAFLD liver and selective export into EVs. miRNAs have emerged as major players in the pathogenesis and progression of NAFLD, with altered hepatic expression of miR-122, -192, and -128-3p disrupting several facets of hepatic lipid metabolism, insulin sensitivity, and cholesterol-lipoprotein trafficking [29,35,47]. Prior studies have also emphasised the pathogenic consequences of vesicular miRNAs from hepatocytes in response to lipotoxic injury in NAFLD [13]. EV-derived miR-192 was recently reported to induce pro-inflammatory polarisation of liver macrophages [30] and EV miR-128-3p internalised by hepatic stellate cells led to marked fibrogenic activation [48]. Decreased cellular miR-122 abundance has been observed in conjunction with increased abundance in circulating EVs. This observation has been attributed to accelerated miR-122 export via the exosomal pathways resulting from metabolic and ER stress in hepatocytes [49,50]. Indeed, multiple reports [51,52,53] support the notion that miR-122 increases specifically in the circulating vesicle fraction in NAFLD, while other liver pathologies (e.g., drug-induced liver damage) are better characterised by changes in the protein-associated fraction [12,53]. Further studies are required to elucidate the intracellular source of miRNAs that are packaged into EVs and the specific mechanisms of export during NAFLD and normal physiology. Nonetheless, the present findings are promising with respect to increasing specificity by targeting analyses to liver-derived miRNA expression.

It is acknowledged that additional clinical data regarding the patient cohort would have been useful and that the inability to access these data is a limitation of the current study. Specifically, NAFL and NASH are hepatic manifestations of metabolic syndrome and are closely associated with multiple comorbidities including type 2 diabetes and obesity, which if unaccounted for, may confound biomarker screening. By way of example, while the miRNA markers evaluated here have previously been associated with NAFL and NASH [13], the current study is unable to definitively exclude that these markers are specific to the hepatic manifestations of the syndrome and not alternative comorbidities observed in this patient population. Additionally, it is acknowledged that, while the purity of ASGR1+ EVs in control EV preparations (Appendix A and Appendix A) was extensively characterised, limitations regarding sample volume precluded comparable analyses in the NAFL and NASH samples. Future studies confirming the observations reported here will benefit from more extensive clinical data and the capacity to undertake additional characterisation of post immunoprecipitation samples.

To date, studies of cell-free miRNA biomarkers have dichotomised subjects to discriminate controls from subjects with either NAFL or NASH or simply pooled as NAFLD. As summarised in our recent review [13], ROC analysis has identified strong predictive performance of several miRNA markers quantified from total cell-free RNA [13]. Up to 7-fold increases in circulating miR-122 and -192 has been previously reported in NAFLD patients [54,55]; however, such significant dysregulation in total cfRNA was not observed in comparison to controls in this cohort. For a number of group pairings, however, ROC analysis produced similarly strong discrimination of cohorts, as seen previously [13]. However, in order to investigate the capacity for miRNA biomarkers from different sources (ASGR1+ EV, global EV, and total cfRNA), to predict an individual’s disease status across control, NAFL, and NASH groups simultaneously, ordinal logistic regression was applied. With this more robust approach that incorporated the biological link of increased biomarker expression in a directional manner, ASGR1+ EVs consistently exhibited excellent diagnostic accuracy. For miR-122 and -192, statistically significant associations were achieved only by the analysis of liver-specific EV miRNA.

## 5. Conclusions

The findings presented here represent the first direct evidence supporting the hypothesis that tissue-specific isolation enhances predictive performance of EV-derived miRNA biomarkers. This study further provides the necessary foundational evidence demonstrating that liver-derived EVs represent a promising solution to current shortcomings with regard to the clinical management of NAFLD. Specifically, this study determined that refining the source from which circulating miR biomarkers are analysed enhanced their capacity to distinguish NAFLD patients from controls. While total cfRNA has long dominated the focus of this field, the belief that markers obtained from a particular blood compartment may better represent disease-associated changes has emerged with the promise of improved biomarker specificity and reproducibility. Here, we show that the selective isolation of liver-specific EVs by anti-ASGR1 immunoprecipitation has the potential to elucidate useful trends in miR expression, which may be applied to track NAFLD patients across the spectrum of clinical disease. This approach not only facilitates the use of biomarkers with ubiquitous expression in unrelated tissues, which may otherwise limit their utility, but also opens the possibility of discovering other highly disease-specific biomarkers, such as EV-derived proteins, from just the affected organ. Such tools can facilitate early diagnosis, identification of patients at greatest risk of progression, serial sampling, and longitudinal monitoring. Thus, the development of liver-EV biomarkers will improve patient management universally, with profound impacts on clinical practice, utilisation of healthcare resources, and advancement of therapeutics.

## Figures and Tables

**Figure 1 biomedicines-10-00195-f001:**
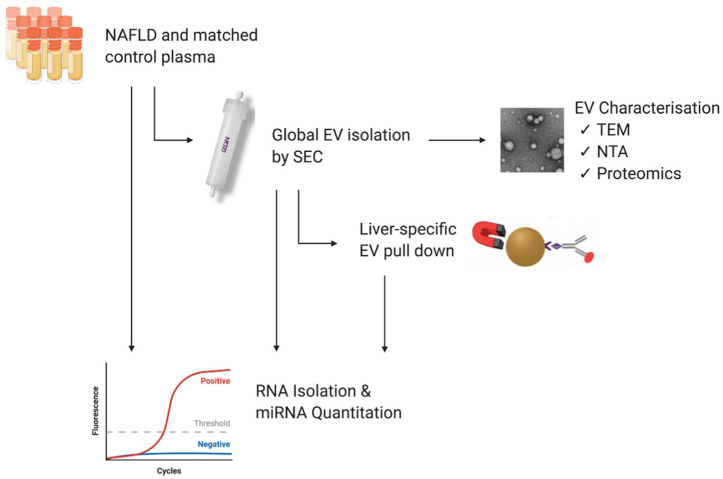
Study workflow. Plasma samples from patients with non-alcoholic fatty liver disease and matched healthy controls were purchased from Discovery Life Sciences (DLS). Samples were aliquoted for miRNA quantitation directly from plasma and from EVs following their isolation by qEV size exclusion chromatography (SEC) and immunoprecipitation (IP). EVs isolated by qEV were characterised by transmission electron microscopy (TEM), nanoparticle tracking analysis (NTA), and protein expression. Figure was created using BioRender.com [https://app.biorender.com/; accessed 17 January 2022].

**Figure 2 biomedicines-10-00195-f002:**
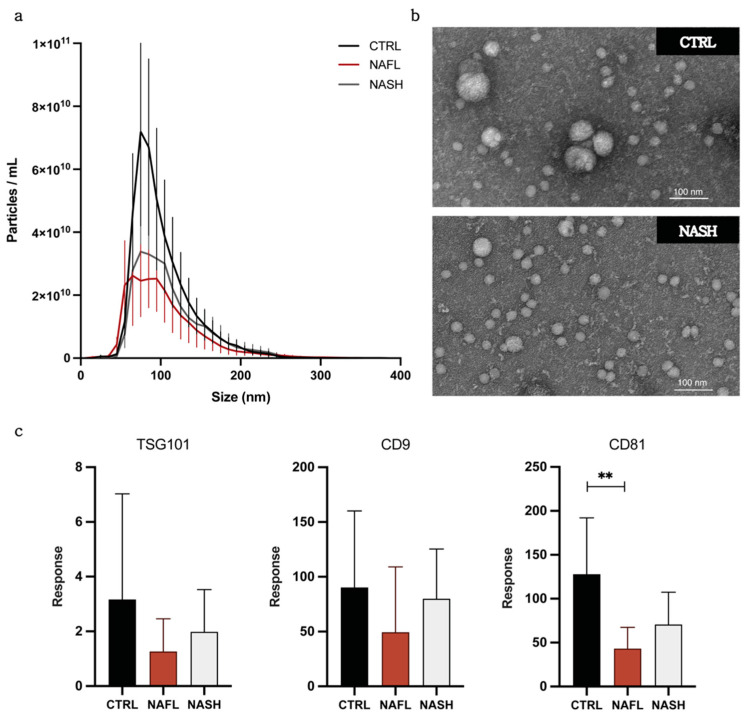
Characterisation of global circulating EV isolated from control, NAFL, and NASH subjects. (**a**) Particle concentration and size distribution by nanoparticle tracking analysis (NTA) (*n* = 5). Error bars denote SEM. (**b**) Representative TEM images of NASH patient and control global EVs. (**c**) Relative abundance of EV protein markers determined by mass spectrometry. Error bars denote SD. ** *p* ≤ 0.01.

**Figure 3 biomedicines-10-00195-f003:**
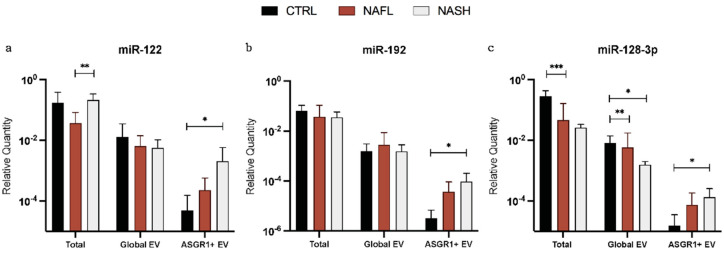
Differential expression of miRNA biomarkers in NAFLD. Relative quantities of miR-122 (**a**), miR-192 (**b**), and miR-128-3p (**c**) normalised to cel-miR-54 in total circulating RNA, global EVs, and asialoglycoprotein receptor 1 (ASGR1) positive EVs isolated from patients with NAFL and NASH and from controls. Statistical analysis performed by the Kruskal-Wallis test with Dunn’s for multiple comparisons. * *p* ≤ 0.05, ** *p* ≤ 0.01, *** *p* ≤ 0.001. Error bars represent standard deviation.

**Figure 4 biomedicines-10-00195-f004:**
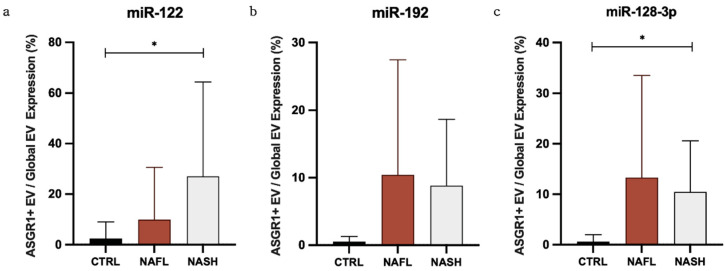
Relative expression of miRNA in liver-specific EVs. Expressions of miR-122 (**a**), miR-192 (**b**), and miR-128-3p (**c**) in asialoglycoprotein receptor 1 positive (ASGR1+) EVs as a percentage of global EVs in NAFL and NASH patients compared with controls. Statistical analysis performed by the Kruskal–Wallis test with Dunn’s for multiple comparisons. * *p* ≤ 0.05. Error bars represent standard deviation.

**Figure 5 biomedicines-10-00195-f005:**
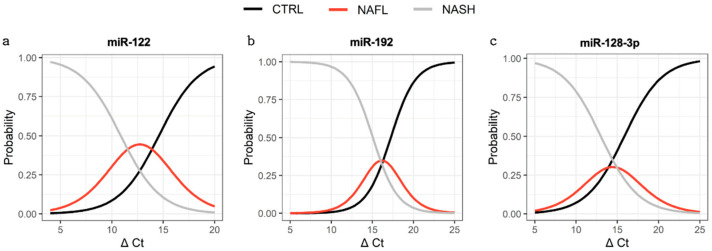
Ordinal logistic regression models. Ordinal regression applied to distinguish control, NAFL and NASH subjects using ASGR1+ EV-derived miR-122 (**a**), miR-192 (**b**), and miR-128-3p (**c**).

**Table 1 biomedicines-10-00195-t001:** Demographic information for control, non-alcoholic fatty liver (NAFL) and non-alcoholic steatohepatitis (NASH) study populations.

Characteristic		Control (*n* = 14)	NAFL (*n* = 8)	NASH (*n* = 6)
Age	Mean (±SD) years	46.5 (15.7)	48.7 (17.7)	53.2 (15.4)
Sex	Female (%)	42.9	57.1	50.0
Race	Caucasian (%)	78.6	57.1	83.3
	Other (%)	21.4	14.3	16.7
	Unknown (%)	0	28.6	0

**Table 2 biomedicines-10-00195-t002:** Mean concentration and size of particles in qEV isolates from control, NAFL, and NASH subjects determined by nanoparticle tracking analysis.

Group (*n* = 5)	Concentration (Particles/mL)	Mean Size (nm)
Control	4.17 × 10^11^ ± 1.76 × 10^11^	102.9 ± 2.7
NAFL	2.34 × 10^11^ ± 9.03 × 10^10^	113.3 ± 10.4
NASH	2.73 × 10^11^ ± 1.01 × 10^11^	110.1 ± 8.6

**Table 3 biomedicines-10-00195-t003:** Expression of miRNA biomarkers in global EVs as a percentage of total circulating RNA and in asialoglycoprotein receptor 1 positive (ASGR1+) EVs as a percentage of global EVs from control, NAFL, and NASH subjects.

Expression (%)		miR-122		miR-192		miR-128-3p	
		Mean	SD	Mean	SD	Mean	SD
Global EV/Total	Control	5.6	10.0	5.1	14.5	2.3	3.7
	NAFL	23.3	27.3	8.7	6.7	17.0	26.2
	NASH	5.1	10.6	3.0	3.4	2.6	1.0
ASGR1+ EV/Global EV	Control	2.4	6.6	0.5	0.8	0.6	1.4
	NAFL	9.9	20.7	10.4	17.0	13.3	20.2
	NASH	27.1	37.2	8.8	9.8	10.5	10.1

**Table 4 biomedicines-10-00195-t004:** Area under the receiver operator characteristic curve for miRNA biomarkers isolated from total cell-free RNA, global EVs, and ASGR1+ EVs.

Source	All NAFLD—CTRL					
	miR-122		miR-192		miR-128-3p	
	AUC	*p*	AUC	*p*	AUC	*p*
Total RNA	0.607	0.335	0.714	0.054 *	0.924	0.0001 *
Global EV	0.505	0.963	0.582	0.462	0.888	0.001 *
ASGR1+ EV	0.830	0.004*	0.895	0.003 *	0.803	0.014 *

* denotes significant *p*-values (≤0.05).

## Data Availability

The data that support the findings of this study are available from the corresponding author upon reasonable request.

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
