# Peer review of "Selective Isolation of Liver-Derived Extracellular Vesicles Redefines Performance of miRNA Biomarkers for Non-Alcoholic Fatty Liver Disease"

_biomedicines, 2022, doi:10.3390/biomedicines10010195_

Round 1

Reviewer 1 Report

In the present manuscript, Newman et al. investigated the predictive performance of miR -122, -192 and -128-3p to distinguish 3 categories of patients: “control”, NAFL and NASH. They measured the expression of these miRNAs in total cell-free serum, purified extracellular vesicles (EVs) and liver-specific EVs. They calculated the correlation and the predictive value of each miRNA and concluded that measuring miRNAs in liver-specific EVs improves the capacity to distinguish NAFLD patients from controls compared to global EVs or cell-free serum. Despite the study is very interesting, there are several concerns that should be addressed by the authors.

  • Are all the experiments presented in the manuscript performed on all the patients (n=14+6+8)? It is not clearly stated for each analysis. Moreover, M&M indicates NAFL n=8 & NASH n=6 but the table indicates NAFL n=6 and NASH n=8.
  • Table 1. Patient characterization is poor. As stated by authors, NAFL and NASH are commonly associated with obesity and type 2 diabetes. Are patients (control, NAFL and NASH) matched for obesity (BMI and waist circumference) and type 2 diabetes (glycemia, HbA1c, 2h OGTT)? All these parameters should be provided in table 1.
  • EV characterization (number and size) was performed on global EVs for the 3 groups of patients and revealed no major difference. However, liver-specific EVs, which are the main focus of the manuscript, were not. What about the concentration and size of liver-specific EVs between the 3 groups? This is a critical point given that previous works suggested that liver-specific EVs increase with NAFL/NASH and that it can completely explain the increase of the 3 miRNAs observed by the authors.
  • Table S1 & Fig S3. It seems that the control of the liver-specific EV purity was only performed on control samples. What about NAFL or NASH? Do authors have the same purity?
  • For valid reasons, authors chose to not normalize the qPCR based on a housekeeping gene and to use a define volume for all the samples. Therefore, the relative quantity of miRNA in sample is mainly dependent on the concentration of EVs and the amount of miRNA in EVs. For global EVs, they measured the concentration and showed no major differences between the 3 groups. However, as explained above, it was not performed for liver-specific EVs. In the current situation, it is not possible to conclude that measuring any of the 3 miRNAs in the purified liver-specific EVs is more interesting than simply quantifying the number of liver-specific EVs in blood (much easier to implement as biomarker). Indeed, the 3 miRNA followed exactly the same pattern strongly suggesting a higher amount of liver-specific EVs with NAFL and NASH.
  • The number of patients is low (n=6 in one group) to properly assess the ability of a biomarker to discriminate groups of patients. It is therefore not surprising that they cannot discriminate NAFL versus NASH and significance is mainly reached when NAFL and NASH patients are merged together. Inclusion of additional patients should improve this point.

Author Response

 Reviewer 1:
1. Are all the experiments presented in the manuscript performed on all the patients
(n=14+6+8)? It is not clearly stated for each analysis. Moreover, M&M indicates NAFL
n=8 & NASH n=6 but the table indicates NAFL n=6 and NASH n=8.
We thank the Reviewer for noting this error. We have corrected the values in Table 1 to reflect
the sample size for each cohort. In response to the Reviewer’s request, we have included a
statement in materials and methods that all the analyses were performed on all the patients
unless indicated otherwise (line 104). The only exception is for EV characterisation
experiments involving nanoparticle tracking analysis (NTA). NTA was performed on a subset
(n=5 per group) of subjects. This was stated in the figure legend for Figure 2 (A) and we have
added this to Table 2.
2. Table 1. Patient characterization is poor. As stated by authors, NAFL and NASH are
commonly associated with obesity and type 2 diabetes. Are patients (control, NAFL
and NASH) matched for obesity (BMI and waist circumference) and type 2 diabetes
(glycemia, HbA1c, 2h OGTT)? All these parameters should be provided in table 1.
We acknowledge the Reviewer’s statement and have made multiple attempts to access the
additional clinical data requested by the Reviewer. Unfortunately, despite reaching out on
multiple occasions and offering to pay for these additional data the sample vendor has not
been forthcoming with these additional clinical data. As per email correspondence / instruction
from the editorial office, we have acknowledged this limitation in the discussion (line 448) and
proceeded with the rest of the resubmission. We agree that these data would be useful, albeit
not essential, but cannot provide them.
3. EV characterization (number and size) was performed on global EVs for the 3 groups
of patients and revealed no major difference. However, liver-specific EVs, which are
the main focus of the manuscript, were not. What about the concentration and size of
liver-specific EVs between the 3 groups? This is a critical point given that previous
works suggested that liver-specific EVs increase with NAFL/NASH and that it can
completely explain the increase of the 3 miRNAs observed by the authors.
The reviewer makes a valid point here that changes in relative abundance of liver specific EVs
could indeed contribute the increase in observed expression of the 3 miRNA biomarkers.
Given limitations in clinical sample volumes, we did not perform NTA on post IP NAFLD
samples. NTA was performed on a subset of control liver specific EVs to determine the
concentration and size of the vesicles and demonstrate the efficiency and recovery of the
immunoprecipitation technique. Fundamentally, this manuscript conveys the message that

selectively isolating this ASGR1+ population improves the ability to use these miRNA markers.
Irrespective of changes in liver-specific EV abundance, we highlight that removing the nondisease-specific background signal is a critical step. This point is reinforced by the lack of
differences in global EV numbers and that increased miRNA expression could not be detected
in global EVs. At a preliminary stage, this work provides justification for the expenditure of
additional efforts to isolate liver-specific EV-derived markers in future studies. Since changes
in miRNA expression may be a result of increased copy numbers packaged into EVs,
increased EV release, or a combination of both – as we discuss in lines 472-475 and 486-496
- in such studies, one could identify whether increased biomarker expression is accompanied
by increased EV release from the liver.
4. Table S1 & Fig S3. It seems that the control of the liver-specific EV purity was only
performed on control samples. What about NAFL or NASH? Do authors have the same
purity?
We acknowledge the Reviewer’s comment, unfortunately due to limitations regarding sample
volume it was not possible to perform all supplemental analyses with the NAFL and NASH
samples in the current study. Given that the purity and capacity to selectively isolate liver EVs
has been reported in our prior publication (see Rodrigues et al 2021 Clin Pharm Ther), other
analyses providing novel findings were given higher priority. We note that prior data generated
on different NAFL samples has demonstrated comparable purity to the analyses reported in
the supplemental data.
5. For valid reasons, authors chose to not normalize the qPCR based on a housekeeping
gene and to use a define volume for all the samples. Therefore, the relative quantity of
miRNA in sample is mainly dependent on the concentration of EVs and the amount of
miRNA in EVs. For global EVs, they measured the concentration and showed no major
differences between the 3 groups. However, as explained above, it was not performed
for liver-specific EVs. In the current situation, it is not possible to conclude that
measuring any of the 3 miRNAs in the purified liver-specific EVs is more interesting
than simply quantifying the number of liver-specific EVs in blood (much easier to
implement as biomarker). Indeed, the 3 miRNA followed exactly the same pattern
strongly suggesting a higher amount of liver-specific EVs with NAFL and NASH.
The Reviewer does make a valid point here regarding possible changes in abundance of liver
specific EVs in individuals with NAFL and NASH. We do however emphasise some limitations
to the Reviewer’s suggestion that simply quantifying the number of liver specific EVs in blood
would be a much easier to implement as biomarker. Specifically, quantification of EV count is
typically achieved by NTA, in the absence of fluorescent labelling NTA simple quantifies the

number of particles of a given size within the sample and does not robustly discriminate EVs
from other particles of comparable size (e.g. lipoproteins). As such NTA is considered a crude
quantification with much lower fidelity than miRNA expression. As we demonstrate here, in
the case of post IP EV isolation miRNA is only derived from EVs and therefore is a far more
robust metric. Furthermore, NTA requires a high level of operator proficiency and is a low
throughput, single sample per analysis technique. For these reasons, while useful as a tool,
NTA is not amenable to clinical translation as a biomarker strategy.
6. The number of patients is low (n=6 in one group) to properly assess the ability of a
biomarker to discriminate groups of patients. It is therefore not surprising that they
cannot discriminate NAFL versus NASH and significance is mainly reached when
NAFL and NASH patients are merged together. Inclusion of additional patients should
improve this point.
The Reviewer makes an important point. We have added a sentence in the discussion to
emphasise that the current study was a proof-of-principle and was not powered to definitively
characterise the statistical significance of this difference.

Reviewer 2 Report

Nonalcoholic fatty liver disease (NAFLD) is the most common chronic liver disease. The search for non-invasive markers of NAFLD and their high efficiency is an urgent and modern research direction around the world. The authors showed that tissue-specific release increased the prognostic efficiency of EV-derived miRNA biomarkers. The development of liver EV biomarkers will improve the treatment of patients worldwide, which will have profound implications for clinical practice, healthcare resource utilization, and improved therapeutic options. The novelty of this research is the use of modern techniques.
The following are some pointers:
1. It is well known that the pathogenesis of NAFLD is inextricably linked to obesity and type 2 diabetes. Authors must provide detailed information about the patients included in the study: BMI, lipid and carbohydrate metabolism, glucose levels, glycated hemoglobin.
2. The authors analyzed the different stages of NAFLD (NAFL, NASH). There is a need to clarify the criteria for the diagnosis of NAFL, NASH. As a reviewer, I need to be sure that the diagnosis was correct. Images of liver biopsy specimens in study groups (control, NAFL, NASH) should be added. The inclusion and exclusion criteria for this study should be stated.
3. The authors studied 3 groups: control (n = 14), NAFl (n = 6), Nash (n = 8). As a reviewer, I believe that the sample size is insufficient for the analysis of ROC. The authors should comment on this observation.
4. Why was the amount of cel-miR-54 equal to 2.5 femtomoles? The author should explain.

Author Response

 Reviewer 2:
1. It is well known that the pathogenesis of NAFLD is inextricably linked to obesity and
type 2 diabetes. Authors must provide detailed information about the patients included
in the study: BMI, lipid and carbohydrate metabolism, glucose levels, glycated
haemoglobin.
We acknowledge the Reviewer’s statement and have made multiple attempts to access the
additional clinical data requested by the Reviewer. Unfortunately, despite reaching out on
multiple occasions and offering to pay for these additional data the sample vendor has not
been forthcoming with these additional clinical data. As per email correspondence / instruction
from the editorial office, we have acknowledged this limitation in the discussion (line 448) and
proceeded with the rest of the resubmission. We agree that these data would be useful, albeit
not essential, but cannot provide them.
2. The authors analyzed the different stages of NAFLD (NAFL, NASH). There is a need
to clarify the criteria for the diagnosis of NAFL, NASH. As a reviewer, I need to be sure
that the diagnosis was correct. Images of liver biopsy specimens in study groups
(control, NAFL, NASH) should be added. The inclusion and exclusion criteria for this
study should be stated.
We appreciate the reviewer’s recommendation for greater clarity in the definitions of patient
cohorts. As stated in the materials and methods, NASH samples are all biopsy proven. For
ethical reasons, biopsy specimens are not available for healthy controls. Similarly, this is not
available for NAFL diagnosis as liver biopsy is avoided as much as possible in routine clinical
care. Other non-invasive test results (e.g. ultrasound, FibroScan) are used instead to
determine presence of hepatic steatosis and degree of fibrosis. Only patients with suspected
NASH are subjected to biopsy. Images of liver biopsy specimens for NASH patients were not
available from the providing company (DLS), but inclusion in the sample collection protocol
was contingent on confirmation of diagnosis by a physician. A statement regarding inclusion
(clinical diagnosis by a physician) and exclusion (viral disease) criteria was included in lines
102-103.
3. The authors studied 3 groups: control (n = 14), NAFL (n = 6), Nash (n = 8). As a
reviewer, I believe that the sample size is insufficient for the analysis of ROC. The
authors should comment on this observation.
The Reviewer makes an important point. As per our response to Reviewer 1, we have added
a sentence in the discussion to emphasise that the current study was a proof-of-principle and
was not powered to definitively characterise the statistical significance of this difference.

4. Why was the amount of cel-miR-54 equal to 2.5 femtomoles? The author should
explain.
A known quantity of C. elegans miR-54 synthetic miRNA mimic was spiked into the samples
to be used as an exogenous control. In test samples, the 2.5 fmol amount gave Ct values in
an appropriate range relative to levels of the endogenous miRNA of interest. This technique
has been consistently reported as an effective normalisation strategy in the absence of
appropriate endogenous house-keeping genes, particularly for circulating miRNA analysis.

Round 2

Reviewer 1 Report

Authors did not improve the manuscript. Reviewer pointed out several issues but no modification was made. When authors are aware of limitations to their study (bias of obesity/T2D, no characterization of liver-specific EVs in the 3 groups, no purity evaluated in the group of interest (NAFL or NASH), …), it is expected than authors rule out these limitations by doing experiments or, at least, moderate their conclusions.

-Reviewer understand the impossibility of having some information such as the clinical parameters. For sake of clarity for readers, authors should clearly state about this limitation in the discussion, unlike done in the revised version. They should explicitly develop that it cannot be excluded that the identified EV-derived miRNA biomarkers are better associated with obesity or T2D than NAFLD.

-About purity: Authors wrote in rebuttal: “We note that prior data generated on different NAFL samples has demonstrated comparable purity to the analyses reported in the supplemental data.” Why they are not showing it if they have the data?

Author Response

 Reviewer 1
When authors are aware of limitations to their study (bias of obesity/T2D, no characterization
of liver-specific EVs in the 3 groups, no purity evaluated in the group of interest (NAFL or
NASH), it is expected than authors rule out these limitations by doing experiments or, at
least, moderate their conclusions.
We acknowledge the Reviewer’s perspective regarding the requested additional caveats and
modifications to the discussion. We specifically agree with the comments regarding the value
of additional clinical data, which we have made every attempt to collect for the samples used
in the current analysis. As suggested, we have made further changes to the discussion
(tracked throughout) to more thoroughly address the limitations cited by the Reviewer.
Reviewer understand the impossibility of having some information such as the clinical
parameters. For sake of clarity for readers, authors should clearly state about this limitation in
the discussion, unlike done in the revised version. They should explicitly develop that it
cannot be excluded that the identified EV-derived miRNA biomarkers are better associated
with obesity or T2D than NAFLD.
As requested, we have added a new paragraph “It is acknowledged that additional clinical
data regarding the patient cohort would have been useful in this regard, and that the inability
to access these data is a limitation of the current study. Specifically, NAFLD and NASH are
hepatic manifestations of metabolic syndrome and are closely associated with multiple
comorbidities including type 2 diabetes and obesity, which if unaccounted for may confound
biomarker screening. By way of example, while the miRNA markers evaluated here have
previously been associated with NAFLD and NASH, the current study is unable to
definitively exclude that these markers are specific to the hepatic manifestations of the
syndrome, and not alternative comorbidities observed in this patient population.”
About purity: Authors wrote in rebuttal: “We note that prior data generated on different
NAFL samples has demonstrated comparable purity to the analyses reported in the
supplemental data
.” Why they are not showing it if they have the data?
We acknowledge the Reviewer’s perspective regarding the value of additional
characterisation specifically with regard to post-IP samples in the disease cohorts and have
explicitly described this in the discussion as a limitation of the current study. We do note that
the independently determined EV Track EV-Metric score for the manuscript, which reflects
the quality of reporting for control experiment is 67%, which is well above the 2021 average
of 51% for all manuscripts (and <40 % for other clinically focussed manuscripts).
The reason that comparable data in NAFL samples are not shown is simply that these
analyses were performed on an independent set of samples that do not relate to the current
study. Specifically, there are two key issues that preclude inclusion of the data (i) we do not
have ethics or collaborators approval to independently publish those data, and (ii) reporting
those data in the current manuscript may be misinterpreted by readers as suggesting that the
analyses were performed on samples related to the current analysis, which would be
misleading.

Round 3

Reviewer 1 Report

Manuscript was improved